# Occurrence of Calcium Oscillations in Human Spermatozoa Is Based on Spatial Signaling Enzymes Distribution

**DOI:** 10.3390/ijms22158018

**Published:** 2021-07-27

**Authors:** Julia Korobkin, Fedor A. Balabin, Sergey A. Yakovenko, Ekaterina Yu. Simonenko, Anastasia N. Sveshnikova

**Affiliations:** 1Center for Theoretical Problems of Physico-Chemical Pharmacology, Russian Academy of Sciences, 30 Srednyaya Kalitnikovskaya, 109029 Moscow, Russia; juliajessika@gmail.com (J.K.); fa.balabin@physics.msu.ru (F.A.B.); 2National Medical Research Center of Pediatric Hematology, Oncology and Immunology Dmitry Rogachev, 1 Samory Mashela St., 117198 Moscow, Russia; 3Faculty of Physics, Lomonosov Moscow State University, 1/2 Leninskie, 119991 Moscow, Russia; 1124422@mail.ru (S.A.Y.); ksimonenko@inbox.ru (E.Y.S.); 4AltraVita IVF and Gynecology Clinic, 4A Nagornaya, 117186 Moscow, Russia; 5Department of Normal Physiology, Sechenov First Moscow State Medical University, 8/2 Trubetskaya St., 119991 Moscow, Russia

**Keywords:** CatSper, sperm cell, calcium signaling, progesterone, computational modeling, reaction-diffusion system

## Abstract

In human spermatozoa, calcium dynamics control most of fertilization events. Progesterone, present in the female reproductive system, can trigger several types of calcium responses, such as low-frequency oscillations. Here we aimed to identify the mechanisms of progesterone-induced calcium signaling in human spermatozoa. Progesterone-induced activation of fluorophore-loaded spermatozoa was studied by fluorescent microscopy. Two computational models were developed to describe the spermatozoa calcium responses: a homogeneous one based on a system of ordinary differential equations and a three-dimensional one with added space dimensions and diffusion for the cytosolic species. In response to progesterone, three types of calcium responses were observed in human spermatozoa: a single transient rise of calcium concentration in cytosol, a steady elevation, or low-frequency oscillations. The homogenous model provided qualitative description of the oscillatory and the single spike responses, while the three-dimensional model captured the calcium peak shape and the frequency of calcium oscillations. The model analysis demonstrated that an increase in the calcium diffusion coefficient resulted in the disappearance of the calcium oscillations. Additionally, in silico analysis suggested that the spatial distribution of calcium signaling enzymes governs the appearance of calcium oscillations in progesterone-activated human spermatozoa.

## 1. Introduction

A spermatozoon is a male reproductive cell. Successful fertilization requires triggering of several events in spermatozoa. These include the acrosome reaction, an exocytotic event that results in eversion of the acrosomal vesicle [1], and the hyperactivation, a type of sperm motility characterized by an asymmetrical beating pattern of the sperm tail [2,3].

Calcium is an important second messenger regulating many cellular activities, including fertilization, secretion, neurotransmission, and cell migration [4]. When extracellular signals appear, a cell can increase its cytosolic Ca^2+^ concentration hundreds of times in milliseconds [5]. Calcium signaling appears to be in the center of sperm activation [6,7,8,9]. In the 1980s, it was demonstrated that cytosolic calcium rise is necessary for the acrosomal reaction [10]. Sperm cytosolic calcium rise was demonstrated in response to zona pellucida recombinant glycoproteins [11], progesterone [10,12], and other types of stimulations [9]. Moreover, spontaneous calcium oscillations occur in up to 50% of the spermatozoa population [13].

In human sperm, intracellular Ca^2+^ ions play a crucial role in both signal transduction and cell homeostasis. Calcium homeostasis is controlled mainly by the plasma membrane calcium ATPase (PMCA) and the Na^+^/Ca^2+^ exchanger (NCX), and calcium plays a crucial role in activating mitochondrial respiration [14]. Mitochondrial abnormalities found in PMCA-deficient sperm [15] suggest a Ca^2+^ overload due to defective Ca^2+^ extrusion. However, in bull sperm, mitochondrial respiration is not up-regulated by Ca^2+^ release [16], and mitochondrial uncoupling does not significantly affect the calcium response occurring in human sperm [17,18].

Progesterone is an endogenous steroid sex hormone involved in the menstrual cycle, pregnancy, and embryogenesis of humans and other species. It is present in the female reproductive tract in nanomolar concentrations and in the human oocyte corona radiata in micromolar concentrations [3].

Different types of responses to progesterone treatment were reported in mammalian spermatozoa—such as single-peak response [6,19,20,21], steady calcium concentration elevation [19,20,21], or calcium oscillations with a period of hundreds of seconds [21,22]. In human sperm, progesterone can trigger acrosome reaction [1] and hyperactivation as well as promote changes in the tyrosine phosphorylation status [22]. Low concentrations of progesterone (10–100 nM) induce sperm motility and activate tyrosine kinases; higher concentrations (1–10 μM) are required to induce acrosome reaction in human spermatozoa [23]. In murine spermatozoa, a transitory increase in intracellular calcium may promote acrosomal exocytosis [19]. In human spermatozoa generating oscillations, the flagella beat mode alternates in synchrony with the oscillation cycle [17]. Low-frequency calcium oscillations observed in a low percentage of human spermatozoa are known to inhibit acrosome reaction [24]. The mechanism of progesterone-induced calcium signaling is not completely clear, although it is known that progesterone modulates the activity of the CatSper (Cation channel of sperm) voltage-gated Ca^2+^ channel [6], and CatSper is crucial for calcium response [3,25]. In the resting state of the cell, CatSper is inhibited by a plasma membrane lipid, 2-arachidonoylglycerol (2-AG) [26]. Progesterone activates abhydrolase domain-containing protein 2 (ABHD2), an enzyme that cleaves 2-AG into arachidonic acid and glycerol, thus activating CatSper [27].

CatSper activation in spermatozoa induces a calcium signaling pathway, which is similar to those typical for non-excitable somatic cells. It starts with phospholipase C (PLC), which catalyzes inositol-1,4,5-triphosphate (IP_3_) production from a membrane phospholipid, phosphatidylinositol 4,5-bisphosphate (PIP_2_). IP_3_, an important secondary messenger, activates inositol-1,4,5-triphosphate channel-receptors (IP3R) located on the spermatozoon intracellular calcium stores, promoting calcium release into the cytoplasm (Figure 1) [28]. Various phosphatases can promote removal of IP_3_ and suppression of further signaling (Figure 1). However, there are several unique features in sperm calcium signaling. First, one of the most prominent isoforms of PLC in mammalian spermatozoa is PLCδ, activated solely by the rise of cytosolic calcium concentration [29]. It has the most significant activity level in mature cells amongst all PLC isoforms [30]. There is a lot of evidence that PLCδ participates in the progesterone-induced calcium response in human spermatozoa [31]. It may get activated when calcium concentration increases due to CatSper opening [27]. Second, the origin of the calcium stores in human sperm is not the endoplasmic reticulum, such as in most somatic cells, but there are two separate stores, the acrosome and the redundant nuclear envelope (RNE), a derivative of the nuclear envelope [32]. Interestingly, each of them is spatially distant from CatSper localization sites by approximately 5 μm [33].

To decode the signaling network described above, a computational modeling approach should be used. Several computational models of sperm cell calcium signaling [34,35,36,37], including progesterone-induced [7,38], have been previously developed. These models have successfully described calcium oscillations in sperm cells of various species as a result of positive feedback from calcium on different plasma membrane channels [34,36,37]. Olson et al. have developed a calcium signaling model in response to 8-Br-cAMP and proved the importance of calcium release from the intracellular stores for the calcium response [38], but this model could not describe the calcium oscillations. Li et al. have built a one-dimensional calcium signaling model in response to progesterone [7]. It could not describe the whole range of periods of calcium oscillations in human sperms varying from 40 to 600 s [13,17]. Another homogeneous model by Simons et al. [39] could describe the occurrence of calcium oscillations with a period about 100 s; however, the calcium concentrations in the model were much higher than the experimental ones (about 100–200 nM [40,41]), as were the basal calcium concentrations (50 nM) [42]. There are no studies describing spontaneous calcium oscillations in human sperm with a period of ~500 s. None of those models incorporates the new data on the 2-AG-dependent inhibition of CatSper published by Miller et al. [22].

Here we investigated the mechanisms of progesterone-induced calcium signaling in human spermatozoa using comprehensive computational modeling and single-cell fluorescent microscopy. We showed that the spatial distribution of the signaling enzymes plays an essential role in forming all types of sperm cell calcium responses to progesterone. We theoretically predicted that the distance between the PMCA localization site and calcium stores determines the lag-time of cytosolic calcium response. Additionally, we demonstrated that the rate of calcium diffusion in cytosol influences the period of calcium oscillations, significantly lowering their frequency.

## 2. Results

### 2.1. Human Sperm Cells Show Three Different Types of Calcium Responses to Progesterone

To determine the patterns of human sperm cell responses to progesterone, we performed fluorescent microscopy of calcium fluorophore-loaded single human sperm cells (see Methods and Appendix A). Calcium concentration was measured using a ratiometric dye Fura-RED, which enabled comprehensive measurements of the signal from moving cells. The sperm heads were immobilized on poly-l-lysine, and the tails were motile and out of the focal plane. For single sperms whose heads were not immobilized properly, the measurement was not performed.

Several types of calcium responses were observed in progesterone-treated human spermatozoa: a single transient calcium increase with the mean width of 160 s and standard deviation of 44 s and mean magnitude of 0.5 μM with standard deviation of 0.2 μM followed by a baseline calcium elevation (Figure 2a–c), a low-frequency oscillatory response (Figure 2d–f), and a steady increase in calcium concentration (Figure 2g,h). In some cases, the transient response had a secondary lower calcium increase starting at 100 ± 20 s (Figure 2c) after progesterone stimulation, and the following calcium concentration was slightly elevated in comparison to non-activated cells (Figure 2b,c). In some cases, the main calcium peak was split (Figure 2b). The oscillatory response was of various types (Figure 2d–f), with a period varying from 80 to 300 s. The mean value was 193 s and the standard deviation was 76 s. Therefore, both the transient response and the calcium oscillations demonstrate a medium level of variance. Approximately half of the cells demonstrated single calcium rise response, while a quarter of cells demonstrated low-frequency oscillations. The total count of responsive cells and count of cells demonstrating each type of response are shown in Figure 2i.

### 2.2. The Three-Dimensional but Not the Homogeneous Model Provided a Quantitative Description of the Progesterone-Induced Calcium Response

Two computational models incorporating the main biochemical reactions (Figure 1) were constructed to describe the known signaling events (Figure 3a). The models’ numerical parameters (protein copy numbers per cell, turnover numbers, and Michaelis constants of the enzymes) were taken from the data on human proteins (see Appendix A for details). Both homogeneous and three-dimensional models were constructed as a “virtual” sperm cell. To describe the experimentally observed calcium responses (Figure 2), the following parameters of the models were varied: the maximal permeability of CatSper (V**_CatSper_**) and the diffusion coefficient for calcium (D_ca_). The rationale behind the variation of V**_CatSper_** was that in the sperm sample incubated in capacitating conditions, there are at least two sperm subpopulations [43], which differ in their membrane potential that affects CatSper (cation channel) activity. D_ca_ is affected by the capacity of the cytosolic buffer [44], particularly by buffering proteins such as 80K-H [45]. In turn, buffering capacity is affected by intracellular pH, which differs between the two subpopulations [43].

The homogenous model (0D) provided a qualitative description of both oscillatory (Figure 3b) and single-peak (Figure 3e) responses. In the 0D model, the maximum width of the calcium peak observed under physiologically reasonable calcium concentrations was 40 s, and the maximum period of calcium oscillations was 30 s. Therefore, the homogeneous model could not describe the experimentally observed peak width (160 ± 44 s) (Figure 3g), oscillations peak width (~100 s), and oscillations period (80–300 s) (Figure 3d) under any parameter value (see Appendix A).

The three-dimensional model (3D) produced oscillations with a 250 s period and a 100 s width of each calcium peak (Figure 3c). The single peak response in the 3D model was 130 s wide (Figure 3f). These modeling results corresponded well with the experimental observations (Figure 3d,g).

### 2.3. Calcium Dynamics in Human Spermatozoa Is Strongly Dependent on the Calcium Diffusion Coefficient

Using the 3D model we developed, a theoretical investigation into the type of calcium response’s dependence on the calcium diffusion coefficient was carried out (Figure 4). It showed that the system might switch its type of response upon variation of calcium diffusion coefficient (Figure 4a). For a fixed diffusion coefficient of IP_3_ equal to 10 μm^2^/s, oscillations in the system appeared when the calcium diffusion coefficient was less than 70 μm^2^/s. In order to investigate the impact of the calcium transition between the principal piece and the head, an auxiliary quasi-heterogeneous model, in which cytosol was divided into two compartments, was built (see Appendix A). This model could describe the experimentally observed low-frequency oscillations (Appendix A), and the period of these oscillations was dependent on the rate coefficient for the calcium transition between the two model compartments (Appendix A). Taken together, these data suggested that some time delay in the movement of calcium from the head to the tail is required for the occurrence of low-frequency oscillations.

Interestingly, oscillations were observed in the 3D model for [IP_3_] < 0.017 µM (Figure 4e). For fixed [IP_3_] = 0.013 µM, the period of oscillations and the delay were increasing for lower calcium diffusion coefficients (Figure 4d).

In the main 3D model, the velocity of calcium extrusion by the PMCA appeared to affect the type of response in progesterone-activated human spermatozoa (Figure 4b). The oscillations were observed for PMCA V_max_ values between 0.7 and 19.5 μM/s, and the oscillations period depended in a near-linear fashion on the velocity of calcium extrusion by PMCA (Figure 4b). In the model where PMCA was located both in the head and in the principal piece, the period of calcium oscillations was reduced to 60 s (Figure 4d, purple line). We also investigated the impact of the spatial distribution of PLCδ on the response type (Figure 4c, dashed line). In the main model, where the PLCδ activity was restricted to sperm head and MP, an oscillatory response occurred (Figure 4d, red line). For a uniformly distributed PLCδ activity (Figure 4d), a significant delay in calcium response appeared (Figure 4d, dotted line).

### 2.4. Mechanisms of Spontaneous Oscillations and Calcium Oscillations in Nominally Calcium-Free Medium Revealed by the Model

The heterogeneous model allowed us to take into account the diffusion of IP_3_ and calcium, thus leading to a description of mechanisms for several experimental phenomena.

First, a low frequency of spontaneous calcium oscillations was described previously [13]. The calcium oscillations with a period of ~500 s could be produced by the heterogeneous model (Figure 5a, dotted curve) with the period (*T_sp_*) and the existence of oscillations depending on the calcium diffusion coefficient (Figure 5b). Interestingly, when the distance (*L*) between the sperm’s principal piece and the sperm’s store RNE was greater than 7.5 µm, the oscillations disappeared (Figure 5a, solid curve).

The model could also describe the behavior of the sperm cell calcium upon the addition of nominally calcium-free buffer (Figure 5c). In accordance with data from [17], in response to a calcium-free buffer, the frequency of calcium oscillations does not change, while the calcium baseline concentration was lower, and the amplitude of calcium oscillations was higher than in a medium that contained calcium. Details of the calculations of the calcium flux change are given in Appendix A.

## 3. Discussion

Here we studied the mechanisms of progesterone-induced calcium signaling in human spermatozoa using computational modeling and single-cell microscopy. We observed three types of calcium responses: the single transient calcium peak with 160 ± 44 s width, the low-frequency calcium oscillations with a period of 80–300 s, and some of the cells responded to progesterone with steady calcium elevation. A quantitative description of all calcium response types specified above was performed, and two computational models were constructed based on the experimental data (a homogeneous and a three-dimensional one). Only the three-dimensional model could describe the low-frequency calcium oscillations. This type of response in sperm originated from the delay required for calcium to diffuse from the calcium store RNE located in the sperm head to PMCA localized in the sperm flagellum. Not only the shape but also the type of calcium response is affected by the spatial system configuration. It may change from transient to oscillatory upon calcium diffusion coefficient lowering.

The observed calcium mobilization responses of the sperm cell to progesterone correspond well with previously published data. Single peak parameters correspond to peak parameters observed by other authors, i.e., to those of calcium oscillations [19,20] and steady calcium elevation [19,20,21]. The mechanism of calcium oscillations proposed here is essentially the same as in the previous computational works for human [7,38] and other species cells [36,37]; that is, there is a positive feedback activation loop from cytosolic calcium on the plasma membrane or RNE calcium channels.

In this study, we demonstrated that the shape and the type of calcium response are affected by the spatial system configuration. Therefore, we can conclude that the longer the sperm cell, the less frequently it will undergo spontaneous oscillations (Figure 5a). This result correlates with the fact that the sperm length was found to be positively associated with semen characteristics [46]. Interestingly, in murine spermatozoa, which are significantly longer than human ones [47], the oscillatory response was observed more frequently than in human ones (9% [48] vs. 15% [19]). In addition, in our model, the oscillations frequency was higher for higher diffusion coefficients. This result is in accordance with the data from [17], that in the progesterone-activated spermatozoa heated to 37 °C, the oscillations frequency was almost two times higher than in those activated at room temperature, as diffusion rate is increased with the temperature increase. For higher calcium diffusion rates in the model, we observed a disappearance of calcium oscillations (Figure 4a), that have a direct physiological significance, as the temperature (and diffusion rates) in female reproductive tract (up to 37.2 °C) [49] is higher than in testis (around 33 °C) [50] and calcium oscillations prevent the acrosome reaction [13].

The dependence of the calcium response on the spatial system configuration is in accordance with the computational models of Pages et al. [51] and Lechleiter et al. [52]. They state that the shape of calcium response in acinar cells and *Xenopus laevis* oocytes may change significantly upon coupling with the fluid flow or accounting for calcium diffusion. The system’s geometry also plays an important role in dendritic cells in calcium response [53]. Low-frequency calcium oscillations may also be explained by calcium release delay generated by ER membrane potential [54], and such delay may be generated by diffusion as well. Interestingly, when the quadrilateral distribution of CatSper channels [36] was added to the model, the model response did not change (data not shown).

Approach limitations should also be taken into account. Mass-action law and Hill functions only approximately describe the activity of enzymes; Fick’s diffusion law is also a rough approximation because calcium diffusion in cells is non-linear [55]. Some of the numerical values (for example, a calcium leak coefficient from the RNE) could not be obtained from the literature and had to be estimated. Another possible limitation is that spermatozoa in different functional states may adhere differently to poly-l-lysine. This is discussed in Appendix A.

## 4. Materials and Methods

### 4.1. Reagents

The sources of materials were as follows: calcium-sensitive cell-permeable fluorescent dyes Fura-2-AM, Fura-RED-AM, and Fluo-4-AM were from Molecular Probes (Eugene, OR, USA); progesterone and DMSO from Sigma-Aldrich (St. Louis, MO, USA); AllGrad^®^ Wash buffer and AllGrad^®^ 90% from Lifeglobal (Guilford, CT, USA). AllGrad^®^ Wash buffer exact electrolyte and supplement composition (1.6 mM calcium, 0.22 mM phosphorus, 2.8 mM potassium, 108 mM chloride, 138 mM sodium mM, 0.24 mM magnesium, 0.18 mM glucose, 4.9 mM lactate, 0.24 mM pyruvate, 5% *w*/*v* HSA, gentamicin sulfate, Phenol Red, HEPES, and bicarbonate), was similar to the Global^®^ Total^®^ single-step medium from Lifeglobal (Guilford, CT, USA), as determined in [56].

### 4.2. Sperm Cells Collection

Sperm samples of 10 adult normozoospermic (according to the World Health Organization) donors were used for the study. The Ethics Committee approved this study for the AltraVita IVF Clinic, decision number #6 from 12 January 2020, and it was conducted according to the Helsinki Declaration’s ethical guidelines. Informed consent was obtained from all subjects. All volunteers were adequately informed about the usage of their clinical and biological data for the experiments before giving their consent.

To evaluate the spermatozoa parameters, the methods routinely used in the AltraVita IVF clinic were performed. Concentration and motility assay was done using a Makler chamber, and evaluation of morphological characteristics was performed using preliminarily stained (with methylene blue and cresyl violet) glass according to Kruger’s criteria (the number of normal spermatozoa is 60–150 million/mL with no less than 70% of them being motile (type a + b) and no less than 13% having a normal morphology).

### 4.3. Microscopy

Before analysis, spermatozoa were centrifuged using 2-layered AllGrad Wash and incubated for 3 h in bicarbonate and HEPES-buffered culture medium with 5% *w*/*v* HSA in tightly closed tubes as described in [57] to induce cells’ capacitation. For microscopy experiments, washed spermatozoa were incubated with 2 μM Fura-RED, AM, or Fluo-8 at 37 °C for 45 min. Cells were resuspended in AllGrad Wash buffer to an approximate density of 10^8^ cells/mL. For epifluorescent microscopy, spermatozoa were immobilized on poly-l-lysine-covered glass slides and investigated in flow chambers [58]. An inverted Nikon Eclipse Ti-E microscope equipped with 100×/1.49 NA TIRF oil objective was used. Cell adhesion was controlled in DIC mode, and fluorescence dynamics were observed in epifluorescence mode. 405 nm and 488 nm lasers were applied sequentially to assess calcium-free Fura-RED and calcium-bound Fura-RED fluorescence in spermatozoa. 488 nm laser was applied in the case of Fluo-8 staining.

Nikon NIS-Elements software was used for microscope image acquisition; Fiji [59] was used for image processing.

### 4.4. Intracellular Calcium Measurement

To measure calcium concentration in sperm cells, the ratiometric Fura-RED-AM dye was used. The values of Fura-RED fluorescence were converted to cytosolic calcium concentration using the following formula [60]:(1)$$\left[Ca^{2+}\right]=K_{d}\frac{F_{488max}^{\;}-F_{488back}^{\;}}{F_{488min}^{\;}-F_{488back}^{\;}}\left(\frac{R-R_{min}^{\;}}{R-R_{max}^{\;}}\right)$$
where *K_d_* = 140 nM is the dissociation constant of the calcium-Fura-RED complex [61], *R* is the ratio of fluorescence upon excitation of Fura-RED with λ = 405 nm (calcium-bound dye) to λ = 488 nm (calcium-free dye). *F*_488*max*_ and *F*_488*min*_ are the maximum and minimum possible fluorescence intensities upon excitation with λ = 488 nm, *F*_488*back*_ is the background fluorescence. *R_min_* and *R_max_* are the ratios upon the minimum and maximum possible calcium concentrations. *R_max_* was measured using a 0.025% Triton-X solution in the AllGrad Wash buffer. The maximum ratio detected in Triton-treated sperm cells was 1.9. After that, 5 μM of calcium chelator EGTA was introduced in the solution, and then the cells were incubated for 10 min at room temperature; the minimum ratio detected in spermatozoa was assessed to be 0.2. Sperm cells which exhibited rapid flagella movement were considered viable (Appendix A).

### 4.5. Computational Model Construction and Validation

Two computational models (a homogeneous and a three-dimensional) were developed to study progesterone-induced activation of human spermatozoa.

#### 4.5.1. The Heterogeneous Model of Progesterone-Induced Activation of Human Spermatozoa

The heterogeneous (three-dimensional, 3D) model was constructed automatically in the VCell software [62] based on the 3D structure of a human sperm cell (Appendix A depicts the central 2D slice of the cell) and a set of biochemical reactions underlying cell activation (Figure 1). The fluxes for the biochemical reactions were calculated based on laws of chemical kinetics (either mass action law, Henry–Michaelis–Menthen kinetics, or Hill functions). The parameters of the equations were taken from the literature on human proteins (Appendix A). The initial conditions were devised from the copy numbers of sperm proteins, those were estimated using pax-db protein abundance database [63] for human testis or mice sperm cells and data on total cell protein number (Appendix A) [64]. The detailed model description, including model reactions and parameters, can be found in Appendix A. The full model is a set of 19 differential equations and 52 parameters, with 47 parameters obtained from the literature.

The model sperm cell 3D structure (called “Geometry”) was constructed in the following manner. The Geometry of the model consisted of three membrane compartments: extracellular volume, cytosol, and RNE. The volume and dimensions of the regions were estimated from the sperm microscopy data [32,46,65,66,67]; the details of the process are given in Appendix A, Appendix A. The geometry of the three-dimensional model consists of extracellular space (300 µm^3^); sperm cytosol (18.4 µm^3^) divided into sperm tail (0.4 μm in diameter [65], 46 μm long [46]), sperm MP (0.8 μm in diameter [65], 7 μm long [66]) and sperm head (4.5 × 3 × 1.5 μm [66]); and sperm calcium store RNE [32] (0.11 µm^3^) located between sperm head and sperm MP. The diffusion coefficients and initial concentrations are given in Appendix A. Estimation of the five unknown parameters (Appendix A) was conducted manually. A detailed description of parameter estimation is given in Appendix A.

The heterogeneous model was constructed in a modular fashion; namely, two modules (Figure 1) were constructed, and their parameters were estimated independently, and then the modules were merged and validated against sperm cell data. The first module consisted of the CatSper channel, its activators and inhibitors. The module simulated the CatSper opening dynamics upon the treatment of spermatozoa with progesterone. The module included the CatSper channel protein, alpha-beta-hydrolase 2 (ABHD2), 2-arachidonoylglycerol, and arachidonic acid (AA). The localization of CatSper and ABHD2 was restricted to the PM area denoting the sperm cilia membrane, while the 2-arachidonoylglycerol was uniformly distributed in the plasma membrane. ABHD2 gets activated upon binding to a progesterone molecule from the extracellular space (it can be described by the mass-action law, Appendix A). 2-AG molecules, which inhibit CatSper, get cleaved into glycerol and AA by activated ABHD2, promoting this calcium channel activity (described by the Michaelis–Menthen equation). ABHD2 activity was assumed to be described by Michaelis–Menten kinetics. The CatSper open state probability determined the flux of calcium ions through open CatSper channel into the sperm cytoplasm.

The second module (Calcium module, Figure 1) was focused on sperm cytosolic and store calcium ion concentration and IP_3_ diffusion and concentration dynamics. IP_3_ was considered to be synthesized by PLCδ and further dephosphorylated by IP3-5-phosphatase [68,69].

The PLCδ activity was considered to be dependent on calcium [29], while IP_3_ to activate IPP5P. See Appendix A for equations. The equations governing calcium concentration were similar to those in previously published models, except for the consideration of RNE calcium dynamics and use of a different IP3R model [70], and are given in Appendix A. The diffusion of IP_3_, calcium, and calcium buffers was taken into account using Fick’s law. Some of the reactions were restricted to spermatozoa’s specific sites (Figure 3a, details are given in Appendix A); for example, IP_3_ generation by PLC occurred only in sperm MP and head. This assumption is based on PIP_2_ concentration being significantly lower in the principal piece due to voltage-sensitive phosphatase activity [71].

#### 4.5.2. The Homogeneous Model of Progesterone-Induced Activation of Human Spermatozoa

The homogeneous (zero-dimensional, 0D) model of the sperm cell was based on the assumption of well-mixing of the proteins in all compartments of the model. Therefore, the heterogeneous model differed from the homogeneous model because it took into account the diffusion of calcium and IP_3_. The homogeneous model was constructed automatically in the COPASI software [72] based on the volumes of the compartments of the sperm cell, calculated for the 3D model (Appendix A) and the same set of the biochemical reactions underlying cell activation as the heterogeneous model (Figure 1). The details of the homogeneous model construction are given in Appendix A.

#### 4.5.3. Considerations about the Inclusion of PLCδ into the Models

Human PLCδ catalytic parameters have not been characterized so far. Human sperm cells contain PLCδ [31]. Additional evidence for PLCδ presence and functioning in human sperm cells could be found in Appendix A. To prove the validity of PLCδ (with murine parameters) inclusion in the model, we performed the alignment of the amino acid sequences of murine and human PLCδ by means of BLAST protein alignment (https://blast.ncbi.nlm.nih.gov/Blast.cgi, accessed on 15 October 2020), which demonstrated that these proteins are 80–90% similar; therefore, their kinetic parameters should not differ much (Appendix A). Additionally, we performed sensitivity analysis [73] for the parameters of the PLCδ in the homogeneous model and received scaled sensitivities 0.0002 for K_PLC_ and −2 × 10^−9^ for V_PLC_ (parameter values are given in Appendix A). This indicates that the possible inaccuracy of parameter estimation would not affect the results of the modeling. Additionally, the variation of the PLCδ parameters in the heterogeneous model (Appendix A) also did not have qualitative effects on the results, and the low-frequency calcium oscillations could be observed for fixed IP3 concentrations (Figure 4E), which correspond to the case when a phospholipase C with low calcium sensitivity (for example, PLCγ) is activated upon addition of progesterone. The acquired results suggest that the model behavior is independent of PLC parameters.

#### 4.5.4. Integration of the Models

The set of partial differential equations was constructed in the VCell software [62] based on the scheme of biochemical reactions (Figure 1). This set with the initial variable values (Appendix A) was integrated using the Method of Lines and CVODE in VCell [62]. The corresponding Virtual Cell Model, SpermCalcium, is available in the public domain at http://www.vcell.org/ (accessed on 17 October 2020) under the shared username “Juliajessica”. The homogeneous model was integrated with COPASI [72].

## Figures and Tables

**Figure 1 ijms-22-08018-f001:**
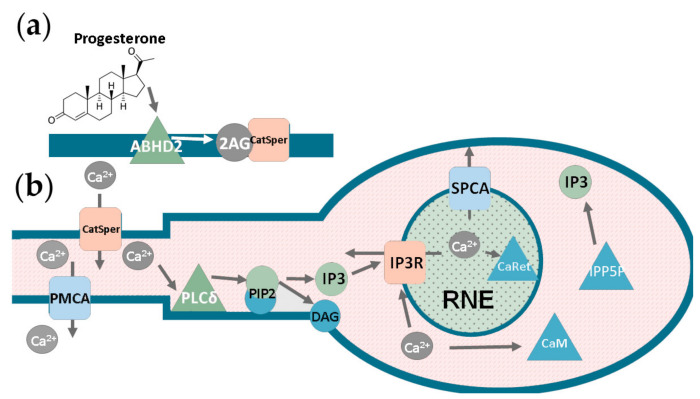
Scheme of the main biochemical reactions during sperm cell activation by progesterone. We suggest a division of the reactions into two modules—the CatSper module and the Calcium module. (**a**) CatSper module. Without activation, the CatSper channel is inhibited by the plasma membrane lipid 2-arachidonoylglycerol (2-AG). Progesterone activates an abhydrolase domain-containing protein 2 (ABHD2), an enzyme that cleaves 2-AG into arachidonic acid and glycerol, and thus activates CatSper. (**b**) Calcium module. Calcium ions enter cytosol through open CatSper channels and then diffuse into spermatozoon midpiece (MP), where a specific isoform of phospholipase C, PLCδ, is located. This enzyme activity is upregulated by calcium; PLCδ catalyzes IP_3_ production from the membrane phospholipid phosphatidylinositol 4,5-bisphosphate. IP_3_ activates IP_3_ channel-receptors (IP3R) located on the spermatozoon calcium store, RNE (redundant nuclear envelope), and thus induces calcium release into the cell cytosol. Plasma Membrane Calcium ATPase (PMCA) present in the sperm cilia membrane extrudes calcium into extracellular media. Signaling Pathway Calcium ATPase (SPCA) pumps calcium into the calcium stores RNE. Moreover, calcium buffers such as calmodulin (CaM) located in the cytosol and calreticulin (CalRet) located in the endoplasmic reticulum can reversibly bind calcium. IP_3_ is metabolized by inositol polyphosphate 5-phosphatase (IPP5P).

**Figure 2 ijms-22-08018-f002:**
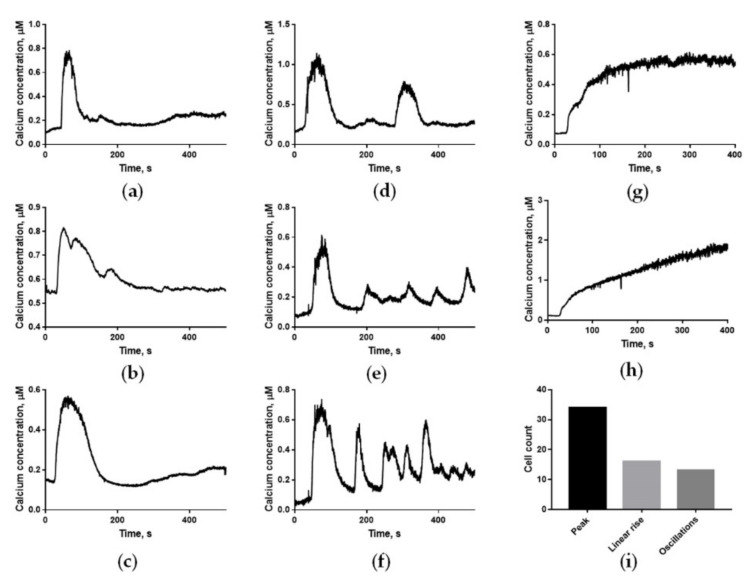
Responses of calcium in human spermatozoa to progesterone at 5 μM. (**a**–**c**) A single transient peak of calcium concentration, 160 ± 44 s wide and 0.5 ± 0.2 μM high, was observed in most progesterone-activated spermatozoa. Sometimes the main peak was followed by a minor calcium elevation (**c**) starting in ~100 s after progesterone activation. The peak width and amplitude were typical for 36 out of 64 cells. (**d**–**f**) Different examples of low-frequency (T~100 s) calcium oscillations detected in progesterone-activated spermatozoa. The oscillation amplitude and frequency were typical for 18 out of 64 cells. (**g**,**h**) A steady calcium increase was observed in some progesterone-treated cells. This type of response was observed in 12 out of 64 cells. (**i**) The histogram of different types of calcium responses in human spermatozoa shows the diversity of calcium responses to activation, with no particular dominating group.

**Figure 3 ijms-22-08018-f003:**
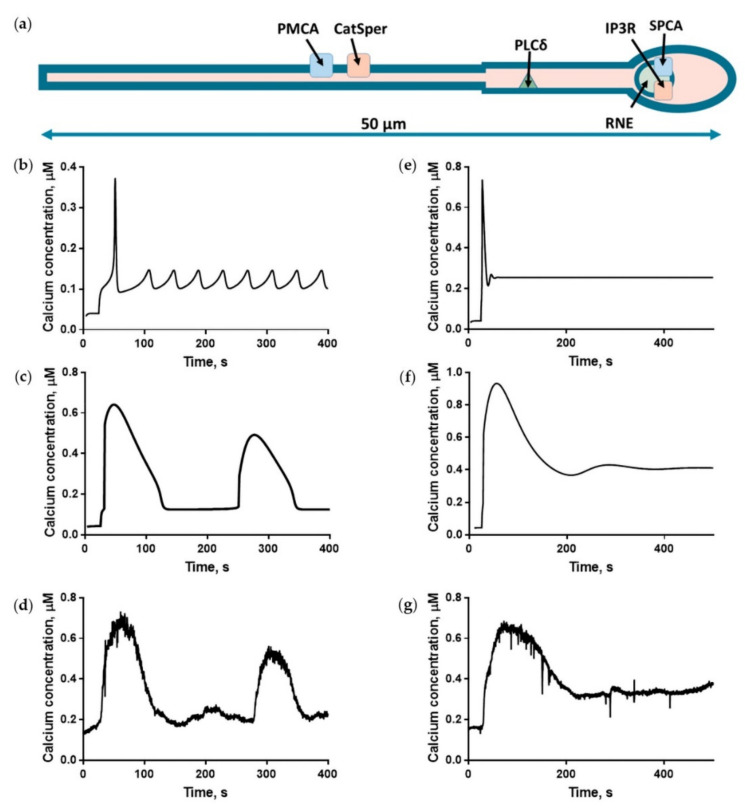
Computational models design and typical simulation results in comparison with experimental data. Progesterone concentration for both the model and the experiment was set at 5 µM. (**a**) Three-dimensional model design. PMCA, as well as CatSper protein location, is restricted to sperm cilia membrane; the corresponding reactions occur only in the specified location. IP_3_ production by PLCδ is also assumed to be happening only in the sperm MP and head. (**b**) Oscillatory response in the homogenous model. The oscillation period (30 s) described by the homogeneous model is significantly lower than the experimentally observed one (100 s). CatSper maximum conductance V**_CatSper_** was set to 0.20 µM/s. Progesterone concentration was set to 5 µM at t = 25 s. (**c**) Oscillatory response in the 3D model. CatSper maximum conductance V**_CatSper_** was set to 0.7 µM/s. D_ca_ was set to 18 µm^2^/s. Progesterone concentration was set to 5 µM at t = 25 s. The model oscillatory response period (200 s) was close to the experimentally observed one. (**d**) Experimentally observed calcium oscillations. (**e**) Homogeneous model single-spike response. The peak width in the homogeneous model (40 s) is significantly lower than that observed in the experiment (160 s). CatSper maximum conductance V**_CatSper_** was set equal to 0.5 µM/s. Progesterone concentration was set to 5 µM at t = 25 s. (**f**) Three-dimensional model of a single peak response. The model single peak width (130 s) was close to the experimentally observed one (160 s). CatSper maximum conductance V**_CatSper_** was set to 1.6 µM/s. D_ca_ was set to 23.5 µm^2^/s. Progesterone concentration was set from 0 to 5 µM at t = 25 s. (**g**) An experimentally observed single peak (see details of the experiments in Figure 2 legend and Methods).

**Figure 4 ijms-22-08018-f004:**
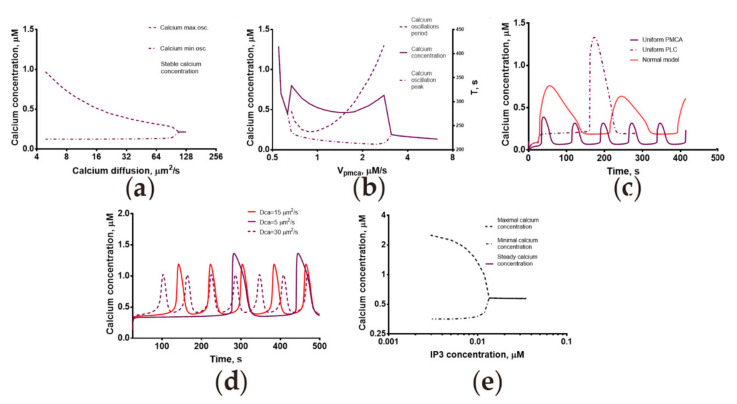
Influence of model parameters on the calcium oscillation response in the 3D model. (**a**) Variation in the calcium ion diffusion coefficient led to the disappearance of the oscillation at higher D_ca_ values. The solid line represents steady-state calcium concentration. The dashed and dotted line represents the lowest concentration upon calcium oscillation). The dashed line represents the highest concentration upon calcium oscillation. (**b**) Variation of the V_max_ for PMCA. The solid black line denotes the steady-state or maximum calcium concentration upon calcium oscillation. The dashed and dotted black line denotes the minimum calcium concentration during oscillations. (**c**) Dependence of the response type on PLCδ distribution. For a PLCδ activity restricted to sperm head and MP, an oscillatory response occurred (red line). For a PLCδ activity distributed uniformly, a significant delay in calcium response appeared (dashed line). For PMCA activity distributed uniformly, the oscillations period was decreased to 60 s (purple line). (**d**) For fixed IP_3_ concentration of 0.013 µm, the oscillations period and the delay were increasing for lower calcium diffusion coefficients. (**e**) Dependence of cytosolic calcium concentration on (fixed) IP_3_ concentration. D_ca_ = 10 µm^2^/s for model plots from (**c**,**d**). V**_CatSper_** = 0.7 µM/s for model plots from (**a**–**e**).

**Figure 5 ijms-22-08018-f005:**
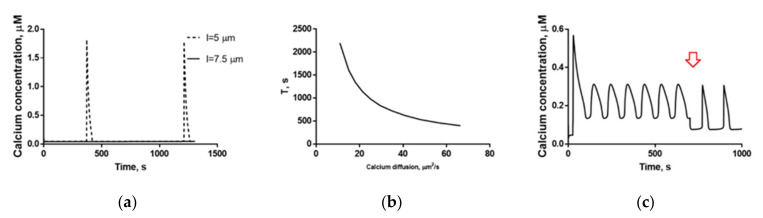
Miscellaneous experimental data described by the heterogeneous model. (**a**) Spontaneous calcium oscillations were demonstrated by the 3D model with distance L between the sperm’s principal piece and sperm’s store RNE of 5 μm (black line), but not 7.5 μm (dashed line). (**b**) Dependence of the model spontaneous oscillations period on the calcium diffusion coefficient D_ca_. (**c**) 3D model calcium oscillations in progesterone-activated spermatozoa upon addition of a nominally calcium-free medium. The moment of setting the conditions to one corresponding to nominally-free calcium medium is shown by the red arrow.

## Data Availability

The datasets generated and/or analyzed during the current study are available in the Google Drive repository, available online https://drive.google.com/drive/folders/1dgNzxcq50eDWuYEwGOzTXQK_bNaDxOUN (accessed on 15 August 2020).

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
