# Peer review of "Occurrence of Calcium Oscillations in Human Spermatozoa Is Based on Spatial Signaling Enzymes Distribution"

_ijms, 2021, doi:10.3390/ijms22158018_

Round 1
Reviewer 1 Report
In the present manuscript authors studied the mechanisms of progesterone-induced calcium signaling using computational modeling and single-cell microscopy. They applied two computational models a system of ordinary differential equations (named homogeneous) and a three-dimensional one (including the variable of space dimensions and diffusion for the cytosolic species). Results showed three types of calcium responses: the single transient calcium peak with 160±44 271 s width; the low-frequency calcium oscillations with a period of 80-300 s; and some of the cells responded to progesterone with steady calcium elevation. Authors stated that the spatial distance between the PMCA localization site and calcium stores affects the rate of the response to calcium stimulus, and that the rate of calcium diffusion in cytosol influences the period of calcium oscillations, significantly lowering their frequency.
This last sentence should be better explored and discussed in the Discussion /Conclusion, especially related to the physiological significance of the study in the comprehension of the sperm functioning.
This is an interesting study, obtained by integrating in silico, empirical and consolidated data. As minor comment, a partial revision of the Discussion better focusing on the impact of their results in the knowledge of sperm physiology, would surely improve the manuscript.
Minor revisions are listed in the attached pdf (especially in the supplementary information, where a general English grammar and style check is required)
Author Response
Thank you for the careful examination of our manuscript and encouraging remarks! We have modified the Discussion section according to your suggestions. Our responses follow your original comments given in bold.
In the present manuscript ... Authors stated that the spatial distance between the PMCA localization site and calcium stores affects the rate of the response to calcium stimulus, and that the rate of calcium diffusion in cytosol influences the period of calcium oscillations, significantly lowering their frequency.
This last sentence should be better explored and discussed in the Discussion /Conclusion, especially related to the physiological significance of the study in the comprehension of the sperm functioning.
Thank you for this suggestion! We have explained in the discussion the significance of sperm cell shape and temperature conditions for physiology.
This is an interesting study, obtained by integrating in silico, empirical and consolidated data. As minor comment, a partial revision of the Discussion better focusing on the impact of their results in the knowledge of sperm physiology, would surely improve the manuscript.
Thank you for the encouragement! We have modified the Discussion accordingly.
Minor revisions are listed in the attached pdf (especially in the supplementary information, where a general English grammar and style check is required)
Unfortunately, no pdf file was attached to the Revision. We have performed additional grammar checks for both manuscript and supplement files.
Reviewer 2 Report
I have read with interest this manuscript. I think that it is potentially interesting but it needs to be improved. in particular I have the following concerns:
- the Author should improve the introduction describing as different biological events the control of calcium clearance in resting conditions, during capacitation and hyperactivation, and during acrosome reaction. They are very different phenomena and must be discussed in the proper way;
the Author should discuss the role of calcium as homeostatic factor and as signal agent: they are different and calcium plays different biological roles in sperm physiology; - the Authors used for their analysis only the sperm cells adherent to the polylysine. Have they considered the hypothesis that only a sperm subpopulation (characterized by a specific biochemical pattern) could be selected? They exist several papers that describe the selective adhesion to different substrates of sperm cells, depending on their functional status.
- the Authors used for their analysis only the sperm cells adherent to the polylysine. Have they considered the hypothesis that only a sperm subpopulation (characterized by a specific biochemical pattern) could be selected? They exist several papers that describe the selective adhesion to different substrates of sperm cells, depending on their functional status.
- the Authors propose different models of calcium oscillation (see Figure 2). But they missed two very important pieces of information: a) the number of sperm cells analysed (total and partial for each single pattern); b) they did not indicate the within-group variability of the measure. Unfortunately, often, sperm cells show a large variability, and it must be considered in analysing any physiological event related with their physiology.
Author Response
Dear reviewer, we are grateful for your evaluation of our work and for insightful remarks! Based on your comments, we have essentially revised the manuscript. Specifically, we have modified the manuscript introduction to address the different aspects of calcium signaling in human spermatozoa, the manuscript title and added the vitality control explanation and several comments on the choice of poly-L-lysine as an adhesion molecule. Our replies follow your original comments given in bold.
the Author should improve the introduction describing as different biological events the control of calcium clearance in resting conditions, during capacitation and hyperactivation, and during acrosome reaction. They are very different phenomena and must be discussed in the proper way; the Author should discuss the role of calcium as homeostatic factor and as signal agent: they are different and calcium plays different biological roles in sperm physiology;
Thank you for your comment! The introduction has been re-written to describe role of calcium in different signaling events and calcium homeostasis
the Authors used for their analysis only the sperm cells adherent to the polylysine. Have they considered the hypothesis that only a sperm subpopulation (characterized by a specific biochemical pattern) could be selected? They exist several papers that describe the selective adhesion to different substrates of sperm cells, depending on their functional status.
Thank you for this valuable remark! Discussion on this topic is added to Approach limitations and S12. It is not clear yet whether the functional state of a sperm cell may influence its adhesion to Poly-L-lysine. However, it is widely used for imaging in human sperm cells, and it is as efficient at binding sperm cells as laminin and is 60% as efficient as simply immersing cells in gelatin or agarose. Even if only a subpopulation of cells is selected, it represents the majority of sperm cells.
the Authors propose different models of calcium oscillation (see Figure 2). But they missed two very important pieces of information:
Thank you for your suggestions! Additional sentences were added to paragraph 2.1. to make a better presentation of the corresponding numbers (within-group variability, number of sperm cells etc).
- a) the number of sperm cells analysed (total and partial for each single pattern);
these numbers are given in Figure 2i and in Section 2.1
- b) they did not indicate the within-group variability of the measure. Unfortunately, often, sperm cells show a large variability, and it must be considered in analysing any physiological event related with their physiology.
The Mean and SD values for each group are given in Section 2.1. We consider the within-group variability to be moderate and the models can describe the observed variability (by variation of V_CatSper and D_Ca).
Reviewer 3 Report
This is a very interesting study relating the calcium oscillations in spermatozoa with the spatial distribution of some signaling enzymes. This study advances a critical field on basic spermatology and the manuscript is well written and structured.
Only some minor comments, and a question: Did you confirmed that analyzed cells were alive and how? You should make this evident. A common failure of many cellular/molecular studies is the inclusion of dead cells, which alter the results.
Title: This study is focused in the human sperm. In fact, the authors make relevant references to studies on other species, and point out the differences among taxons regarding sperm shape and biology. Therefore, the species ("human spermatozoa") should be included in the title.
L49 maybe "in milliseconds" or similar?
L63-64: Revise punctuation?
Fig. 1: No reference to IP3K, shown in the drawing.
L141-143: Shouldn't be better if the explanation followed reading order, left->right?
L147: If "Fig. 2c" correct? (looks like a single peak to me).
L296: °C.
Discussion: Maybe the authors might add their opinion on the relevance of their findings and models for studies on sperm capacitation and fertilization? Calcium oscillations are also relevant on sperm maturation and motility activation upon mixing with seminal plasma.
Author Response
Dear reviewer, thank you for your kind evaluation of our work! We have addressed your questions to the best of our ability. Our replies follow your original comments in bold.
Only some minor comments, and a question: Did you confirmed that analyzed cells were alive and how? You should make this evident. A common failure of many cellular/molecular studies is the inclusion of dead cells, which alter the results.
Thank you for this remark! Analyzed cells were considered viable if we could see their flagella moving rapidly. Fig. S10 has been added to provide an example of such movement.
Title: This study is focused in the human sperm. In fact, the authors make relevant references to studies on other species, and point out the differences among taxons regarding sperm shape and biology. Therefore, the species ("human spermatozoa") should be included in the title.
Thank you! We have modified the title accordingly.
Minor remarks corrected.
Discussion: Maybe the authors might add their opinion on the relevance of their findings and models for studies on sperm capacitation and fertilization? Calcium oscillations are also relevant on sperm maturation and motility activation upon mixing with seminal plasma.
Thank you for your comment! The Discussion section has been updated to describe the possible implications of calcium oscillations regulation by diffusion for the fertilization process.
Round 2
Reviewer 2 Report
I have no further comments